# In Vitro Activity of Essential Oils against *Saprolegnia parasitica*

**DOI:** 10.3390/molecules24071270

**Published:** 2019-04-01

**Authors:** Simona Nardoni, Basma Najar, Baldassare Fronte, Luisa Pistelli, Francesca Mancianti

**Affiliations:** 1Dipartimento di Scienze Veterinarie, Università degli Studi di Pisa, viale delle Piagge 2, 56124 Pisa, Italy; baldassare.fronte@unipi.it (B.F.); francesca.mancianti@unipi.it (F.M.); 2Dipartimento di Farmacia, Università degli Studi di Pisa, via Bonanno 6, 56126 Pisa, Italy; basmanajar@hotmail.fr (B.N.); luisa.pistelli@unipi.it (L.P.)

**Keywords:** *Saprolegnia*, essential oils, in vitro sensitivity, *Thymus vulgaris* L., *Origanum vulgare* L., *Cymbopogon flexuosus* (Nees ex Steud.) Watson, *Litsea cubeba* (Lour.) Pers., *Citrus bergamia* Risso et Poiteau

## Abstract

*Saprolegnia* spp. water molds severely impact fish health in aquaculture, fish farms and hobby fish tanks colonizing mature and immature stages of fishes, as well as eggs. Considering that there are no drugs licensed for treating and/or control the organism, efficient and environmental low-impact methods to control these oomycetes in aquaculture are needed. The aim of the present report was to evaluate the in vitro sensitivity of *Saprolegnia parasitica* to essential oils (EOs) from *Citrus aurantium* L., *Citrus bergamia* Risso et Poiteau, *Citrus limon* Burm. f., *Citrus paradisi* Macfad, *Citrus*
*sinensis* Osbeck, *Cinnamomum zeylanicum* Blume, *Cymbopogon flexuosum* (Nees ex Steud.) Watson, *Foeniculum vulgare* Mill., *Illicium verum* Hook.f., *Litsea cubeba* (Lour.) Pers., *Origanum majorana* L., *Origanum vulgare* L., *Pelargonium graveolens* L’Hér., *Syzygium aromaticum* Merr. & L.M.Perry, and *Thymus vulgaris* L., by microdilution test. The most effective EOs assayed were *T. vulgaris* and *O. vulgare*, followed by *C. flexuosum, L. cubeba* and *C. bergamia.* These EOs could be of interest for controlling *Saprolegnia* infections. Nevertheless, further safety studies are necessary to evaluate if these products could be dispersed in tank waters, or if their use should be limited to aquaculture supplies.

## 1. Introduction

*Saprolegnia* spp. organisms are aquatic oomycetes, known as water molds, which severely impact fish health in aquaculture, fish farms and hobby fish tanks [1]. Saprolegniosis occurs mainly in salmonid aquaculture, but also in wild fish [2]. These oomycetes can colonize mature and immature stages of fishes, as well as eggs [3], causing hyphal growth as cotton-like mycelia on fish body surface and gills. Conversely, fish eggs are killed by hyphae breaking up the chorionic membrane of the eggs, thus provoking an osmotic shock [4]. Fishes are more prone to saprolegniosis in the winter, whilst during the rest of the year, when the water temperature is warmer, the disease is usually less frequent [5].

Up until 2002, the control of these oomycetes in aquaculture was achieved with malachite green, which that year was banned worldwide due to its carcinogenic and toxic effects, causing a re-emergence of *Saprolegnia* infections in aquaculture. To date, no valid substitutes have been found, thus there is a need for identify efficient and environment low-impact methods to control such infections. 

In recent years, the interest for identifying sustainable products from landscape plants has increased and some data are available for *Saprolegnia* spp. species [6,7]. In particular, some essential oils (EOs), mostly from Lamiaceae, have been successfully in vitro and both in vitro and in vivo employed, respectively [8,9,10,11,12]. Some other EOs have been checked with different results [13,14,15,16]. The aim of the present report was to evaluate the in vitro sensitivity of *Saprolegnia parasitica* to some EOs from different plant families.

## 2. Results

### 2.1. GC-MS Analysis

The composition of EOs is reported in Table 1. Tested EOs showed very different chemical composition. Monoterpene hydrocarbons, in particular limonene, were present in amounts near to 90% in *Citrus sinensis* Risso et Poiteau, *Citrus paradisi* Macfad, *Citrus aurantium* L., but not in *Citrus limon* Burm. f. Oxygenated monoterpenes were highly represented in *Cymbopogon flexuosum* (Nees ex Steud.) Watson and *Litsea cubeba* (Lour.) Pers. (neral and geranial), *Origanum majorana* L., *Origanum vulgare* L. (carvacrol), *Pelargonium graveolens* L’Hér. (citronellol), *Thymus vulgaris* L. (thymol). *Citrus bergamia* Risso et Poiteau was characterized by a similar amount of both monoterpene hydrocarbons and oxygenated monoterpenes (49 and 48.4%, respectively). *Illicium verum* Hook.f. ((*E*)-anethol) and *Syzygium aromaticum* Merr. & L.M.Perry (eugenol) were composed by more than 90% of phenylpropanoids. 

### 2.2. In Vitro Assay 

Selected EOs showed different degrees of activity, with MIC values ranging from 0.5% (*T. vulgaris* and *O. vulgare*) to > 10% (*C. sinensis, C. paradisi, C. aurantium, C. limon*). The most effective EOs contained large amounts (52.6%) of thymol (*T. vulgaris*) and carvacrol (65.9% and 20.8%) (*O. vulgare* and *O. majorana*, respectively). More detailed data are reported in Table 2.

## 3. Discussion

The most effective EOs assayed in the present study were *T. vulgaris* and *O. vulgare*. Our data confirm the findings of Perrucci et al. [8], Tampieri et al. [10] and Gormez and Diler [11], who observed a good anti-*Saprolegnia* in vitro activity of these EOs. *O. majorana* showed a MIC value of 1%, resulting an interesting EO against tested oomycetes. At the best of our knowledge, our findings cannot be compared to literature data with regards to the anti-*Saprolegnia* activity of this EO.

Tampieri et al. [10] report in vitro activity of *C. flexuosum* EO. In our study this EO showed a 2.5% MIC value, confirming this result. *L. cubeba* EO, characterized by an analogous chemical composition and never tested before, was therefore included in the assay, demonstrating a 2.5% MIC, also. This same value was obtained testing *C. bergamia*, while further *Citrus* spp. EOs did not show biological activity. *C. limon* EO only has been object of interest by other Authors [10,13] resulting ineffective. The unique efficacy of *C. bergamia* among other species within the genus appears to be very interesting, and it could be explained on the basis of its peculiar chemical composition, characterized by the presence of linalool and linalool acetate, along with a lower amount of limonene, when compared to other *Citrus* species. 

In the present study, rather high MIC values were obtained. *Saprolegnia* oomycetes are characterized by a natural resistance to currently administered antimycotic drugs, also. This fact is probably due to their peculiar cell wall composition, made up of a mix of cellulosic compounds and glycan, and it could have implications for the low sensitivity to EOs commonly very active against organisms referable to Ascomycota and Deuteromycota, frequently involved in human and animal mycoses.

In previous studies performed on different molds, using EOs with similar composition, *T. vulgaris* [20], *O. vulgare* and *L. cubeba* EOs [21], showed marked antifungal activity, with lower MIC values when compared to those obtained in the present work. 

The feasibility of EOs practical application in aquaculture has been prospected by Khosravi et al. [14], who evaluated the activity of natural substances in treating *Saprolegnia*-infected rainbow trout (*Oncorhynchus mykiss*) eggs, referring no statistical difference between in vivo and in vitro results. Despite this fact, they concluded that additional evaluations should be carried out with regard to the toxicity of these natural products. By the way, Hoskonen et al. [12] report the toxicity of clove EO, when used to control saprolegniosis during incubation of rainbow trout eggs. In vivo adverse biological effects, as well as a toxicity of thymol and carvacrol on zebrafish (*Danio rerio*) models, were recently reported by Ran et al. [22] and Polednik et al. [23]. For this reason, the administration of EOs obtained from Lamiaceae family, containing large amounts of phenols, although biologically effective, should be carefully evaluated. Conversely, on the basis of the results obtained in the present study, *L. cubeba*, *C. flexuosum* and *C. bergamia* EOs would represent a promising alternative to above mentioned compounds.

## 4. Materials and Methods

### 4.1. Essential Oils 

For the study, *C. aurantium*, *C. bergamia*, *C. limon*, *C. paradisi*, *C. sinensis* (fam. Rutaceae), *Cinnamomum zeylanicum* Blume (fam. Lauraceae), *C. flexuosum* (fam. Poaceae), *Foeniculum vulgare* Mill. (fam. Apiaceae), *I. verum* (fam. Schisandraceae), *L. cubeba* (fam. Lauraceae), *O. majorana*, *O. vulgare* (fam. Lamiaceae), *P. graveolens* (fam. Geraniaceae), *S. aromaticum* (fam. Myrtaceae) and *T. vulgaris* (fam. Lamiaceae) EOs were employed. All the above mentioned EOs were purchased from Flora srl (Lorenzana, Pisa, Italy). EOs were selected on the basis of literature data, and on the in vitro efficacy previously observed versus yeasts and molds [21,24]. 

### 4.2. GC-MS Analysis

Volatile constituents of each EO were analyzed by GC-MS as previously reported [21]. Briefly, a Varian CP-3800 gas chromatograph equipped with HP-5 capillary column (30 m × 0.25 mm; coating thickness, 0.25 mm) and a Saturn 2000 ion trap mass detector (Varian Inc., Walnut Creek, CA, USA) were employed. Analytical conditions were as follows: injector and transfer line temperature, 220 and 240 °C respectively; oven temperature, programmed from 60 to 240 °Cat 3 °C/min; carrier gas, helium at 1 ml/min; injection, 0.2 ml (10% hexane solution); split ratio, 1:30. Identification of the constituents was based on comparison of the retention times with those of authentic samples, comparing their linear retention indices relative to the series of n-hydrocarbons, and on computer matching against commercial and home-made library mass spectra built up from pure substances and components of known oils and MS literature data [17,18,19]. Amount of EOs constituents was calculated by relative percentage abundance.

### 4.3. In Vitro Assay 

The in vitro antimycotic activity of EOs was evaluated on *Saprolegnia parasitica* isolated from water in a rainbow trout farm and maintained on Malt Extract Broth (MEB). The effectiveness of EOs was assessed by means of a microdilution test carried out using MEB as culture medium. Portions of approximately 1 mm^2^ of mycelium were used as fungal inocula in 96- wells plates (Pbi International, Milano, Italy). One linen seed was put in each well, to enhance fungal growth. Stock solutions at 10% of each chemically defined EO were diluted into MEB to obtain concentrations ranging from 0.1% to 5%, to obtain a minimum inhibition concentration (MIC) value, expressed as the lowest concentration, at which fungal growth was not noticeable. Plates were covered in order to avoid EOs evaporation, and incubated at room temperature for about five days, or until a full development of mycotic growth in control wells was observed. Portions of the inocula, in which growth was not visually noticed, were removed, twice washed in sterile saline, then were seeded onto malt agar plates to evaluate fungal viability. Controls were achieved to evaluate the sterility of each EO as well of the culture medium and seeding fungal inocula in medium without EOs. All tests were performed in quadruplicate. 

## 5. Results

### 5.1. GC-MS Analysis 

The composition of the essential oils (EOs) is reported in Table 1.

### 5.2. In Vitro Assay

Fungal viability test showed a fungicidal activity of EOs characterized by MIC values up to 10%, indicating that MIC and minimum fungicidal concentration were corresponding. Data about the anti-*Saprolegnia* in vitro activity of tested EOs are reported in Table 2.

## 6. Conclusions

The findings of this study indicate that *L. cubeba*, *C. flexuosum* and *C. bergamia* EOs could be of interest for controlling *Saprolegnia.* Further safety studies are needed to evaluate if these products could be dispersed in tank water, or if their use should be limited to aquaculture supplies.

## Figures and Tables

**Table 1 molecules-24-01270-t001:** Chemical composition of tested essential oils (only compounds present at a concentration ≥1% were included).

		Component	LRI ^c^	*LR I^l^*	*C.a*	*C.b*	*C.l*	*C.p*	*C.s*	*C.z*	*C.f*	*F.v*	*I.v*	*L.c*	*O.m*	*O.v*	*P.g*	*S.a*	*T.v*
1	mh	tricyclene	926	928		1.1	1.9			1.4		8.6	1.4	1.5					
2	mh	camphene	953	955							1.1								
3	mh	sabinene	976	976		1.1	2.3							1.0	3.2				
4	mh	β-pinene	980	981		5.4	11.9					1.0		1.2					
5	nt	6-methyl-5-hepten-2-one	985	986							1.8			1.5					
6	mh	myrcene	991	992	1.9	1.0	1.8	2.2	2.3						1.6	2.2			
7	mh	α-phellandrene	1005	1006						2.1		2.2							
8	mh	α-terpinene	1018	1019						1.0					4.7	2.1			
9	mh	*p*-cymene	1026	1026						3.0		1.9			4.2	9.3			15.3
10	mh	limonene	1031	1033	94.7	33.2	65.7	92.2	95.5		2.0	6.5	3.9	16.3	2.1				
11	mh	β-phellandrene	1031	1031						5.9									
12	om	1,8-cineole	1033	1033										2.3					
13	mh	γ-terpinene	1062	1062		6.4	9.3								7.9	5.3			2.9
14	om	*cis-*sabinene hydrate	1068	1074											3.2				
15	om	fenchone	1087	1096								20.1							
16	mh	terpinolene	1088	1089											1.5				
17	om	*trans-*sabinene hydrate	1097	1096											12.8	1.8			3.8
18	om	linalool	1098	1099		14.2				6.3	1.5			1.5			3.9		
19	om	*cis*-rose oxide	1111	1111													1.0		
20	om	menthone	1154	1155													1.1		
21	om	isomenthone	1164	1170													3.5		
22	om	borneol	1165	1165															1.6
23	om	4-terpinenol	1177	1177											17.6				2.4
24	om	α-terpineol	1189	1189											2.7				
25	pp	methyl chavicol (estragole)	1195	1196								8.6							
26	om	citronellol	1228	1128													44.5		
27	om	thymol methyl ether	1232	1235															1.7
28	om	neral	1240	1241							35.2			32.5					
29	om	geraniol	1255	1277							4.4				2.7		13.7		
30	om	linalyl acetate	1257	1257	1.4	31.6									3.2				
31	pp	*(E)*-cinnamaldehyde	1266	1268						56.4									
32	om	geranial	1270	1271			1.2				38.4			36.4					
33	om	citronellyl formate	1280	1295													7.3		
34	pp	*(E)*-anethol	1283	1283								46.9	89.8						
35	om	thymol	1290	1290															52.6
36	om	carvacrol	1298	1298											20.8	65.9			
37	om	geranyl formate	1298	1300													1.9		
38	pp	eugenol	1356	1348						3.0								77.9	
39	sh	α-copaene	1376	1377						1.5									
40	om	geranyl acetate	1383	1381							4.2								
41	sh	*β*-caryophyllene	1418	1418						10.3	2.3				1.7	3.7		8.9	6.8
42	sh	aromadendrene	1439	1440													2.9		
43	sh	α-humulene	1454	1455						2.7									
44	sh	α-muurolene	1499	1499^*^											1.4				
45	sh	*γ*-cadinene	1513	1513							1.2								
46	pp	eugenyl acetate	1524	1525														12.2	
47	sh	*δ*-cadinene	1524	1524															1.0
48	om	citronellyl butyrate	1532	1529													1.2		
49	om	geranyl butyrate	1564	1562^**^													1.1		
50	nt	2-phenylethyl tiglate	1583	1590													1.2		
51	om	citronellyl tiglate	1658	1670													1.2		
52	om	geranyl tiglate	1696	1695													1.2		
53	nt	benzyl benzoate	1762	1750						1.0									
Total of identified compounds					100.0	100.0	100.0	99.2	100.0	99.7	97.6	99.7	100.0	99.4	98.0	98.5	99.3	100.0	95.6
Monoterpene hydrocarbons (mh)					97.4	49.0	94.3	96.2	98.7	15.5	3.9	22.1	7.3	21.3	27.7	22.5			21.5
Oxygenated monoterpenes (om)					1.9	48.4	3.6	0.5	0.6	7.4	86.3	21.1	0.7	75.7	66.5	71.2	85.8		64.2
Sesquiterpene hydrocarbons (sh)					0.2	2.4	2.0	1.9		14.7	4.5		1.1	0.9	3.3	4.2	7.8	9.5	9.2
Oxygenated sesquiterpenes (os)										0.8	0.9		0.1		0.4	0.4	4.5	0.4	
Phenylpropanoids (pp)										1.0		56.5	90.8			0.1		90.1	
Other compounds					0.5	0.1	0.1	0.6	0.7	60.3	2.0			1.5	0.1	0.1	1.2		0.7

LRI ^c^: Linear Retention Index, measured on HP-5 column; LRI ^l^: Linear Retention Index in web book NIST [17] ^*^: Katioti et al. [18]; ^**^: Adams [19]. C.a: *Citrus aurantium*; C.b: *Citrus bergamia*; C.l: *Citrus limon*; C.p: *Citrus paradisi*; C.s: *Citrus sinensis*; C.z: *Cynnamomum zeylanicum*; C.f: *Syzygium aromaticum*; F.v: *Foeniculum vulgare*; I.v: *Illicium verum*; L.c: *Litsea cubeba*; O.m: *Origanum majorana*; O.v: *Origanum vulgare*; P.g: *Pelargonium graveolens*; S.a: *Syzygium aromaticum*; T.v: *Thymus vulgaris*.

**Table 2 molecules-24-01270-t002:** Minimum inhibitory concentrations (MIC) of tested essential oils against *Saprolegnia parasitica.*

Essential oil	MIC (%)
*Citrus aurantium*	>10
*Citrus bergamia*	2.5
*Citrus limon*	>10
*Citrus paradisi*	>10
*Citrus sinensis*	>10
*Cinnamomum zeylanicum*	5
*Cymbopogon flexuosum*	2.5
*Foeniculum vulgare*	5
*Illicium verum*	10
*Litsea cubeba*	2.5
*Origanum majorana*	1
*Origanum vulgare*	0.5
*Pelargonium graveolens*	10
*Syzygium aromaticum*	5
*Thymus vulgaris*	0.5

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
