# Peer review of "In Vitro Activity of Essential Oils against Saprolegnia parasitica"

_molecules, 2019, doi:10.3390/molecules24071270_

Round 1

Reviewer 1 Report

The authors presented the in vitro assessment of essential oil activity against Saprolegnia parasitica in a satisfactory way. The work is interesting, informative and well-written.  I only have some observations that are listed in the following lines.

1.      Line 30: remove “on”.

2.      Line 40: … for identifying...

3.      Lines 41-43: rewrite.

4.      Line 44: …successfully employed both in vitro and in vivo…

5.      Lines 44-45: clarify the statement “Some other EOs have been checked with different results [13-16]”.

6.      Line 51: “Tested EOs, as expected, showed very different chemical composition.” It’s expected to have different composition from different plants. Rewrite or remove.    

7.      Line 56: “Selected EOs checked in the present paper” rewrite.

8.      Lines 110-113: Plant names should be italic.

9.      Line 114: …purchased from….

10.   Remove lines 139-154. They are repeated in the results (from line 39)

11.   Line 156: remove “would”

12.   Rewrite the conclusion.

Author Response

Dear Reviewer,

the response to comments are reported in the attached file.

Very sincerely,

Simona Nardoni and co-Authors 

Reviewer 2 Report

 The subject of this work is not interesting because most of these EOs are well known related to their known volatile chemical constituents and antimicrobial activities. Therefore I think, it is not enough to publish as article.

How produced these EOs, have you any details about the methods of the EOs productions, which geographical sources have these plants. The collection of the plants, the production methods and process should be better to do by authors. This is an essential step for such work.

The names of the plants lack some details: (plant families names and author names of species)

Author Response

Dear Reviewer,

the answers to your concerns are reported in the attached file.

Very sincerely,

Simona Nardoni and co-authors

Reviewer 3 Report

Manuscript presents research on antimycotic activity of fifteen commercial essential oils against Saprolegnia parasitica. It is well structured, but it needs some corrections before publication.

I present my remarks below and mark some of them in the manuscript.

Please list essential oils in all parts of manuscript and Table 2 in the same alphabetic order as in Table 1.

The sentence in ls. 20-21 is misleading because it suggests that Authors of  the manuscript researched in vivo activity of thymol and carvacrol. What is more, it is not clear without reporting the composition of  the most effective EOs. I suggest to omit this sentence in Abstract or rewrite it.

I would like Authors to be aware that the true percentages of EOs components can be obtained by GC with FID detector but not by GC-MS. However, in study presented in manuscript one can accept MS detector. To support the identification both experimental and literature retention indices should be reported in Table 1.

In Table 1 there are many compounds without percentages: 13,14,17,20,24,40,44,51,58,63. Should they be removed?

Some compound names should be corrected both in Table 1 and also in the text: trans- and cis- should be in Italic; between (E) and compound name hyphen should be put, e.g. 36. (E)-cinnamaldehyde.

Other names needing correction both in Table 1 and also in the text are: 28. terpinen-4-ol; 30. methyl chavicol (estragol); 35. linalyl acetate; γ-cadinene; 54. eugenyl acetate; 59. 2-phenylethyl tiglate; 60. citronellyl tiglate or citronellyl angelate. Tiglic acid is trans-2-methylbut-2-enoic acid and its cis isomere is angelic acid.

Compounds 60 and 61 belong to oxygenated monoterpenes. Hence, the percentages of group of compounds should be recalculated. Check please the total percentage of identified compounds in each EO, which for example in first column is 98.0 (not 100) and in the last one 88.1 (not 95.6). There is no need  to report unknown compound in Table 1 unless RI and MS characteristic is given.

Because phenylpropanoids are also non-terpene derivatives it would be better to use "Other compounds" instead of "Non-terpene derivatives" and move this line to the end of Table 1.

Sentence in ls. 55-56 is not true as F. vulgare EO contain 55.6% of phenylpropanoids. In this passage C. bergamia  EO should be mentioned.

Pay attention please that monoterpene thymol and carvacrol are phenols and they are the most effective antimicrobial agents among all essential oil components. In family Lamiaceae there is a lot of EOs containing oxygenated monoterpenes other than phenols as main constituents and general statement in ls. 103-105 should be rewritten.   

 In ls. 139-154 Results are repeated and should be removed.

Explain please what do you mean by "could be released directly in tank water" (l. 23 and 158).

All Latin plant and other organisms names should be in Italic both in the text (e.g. ls. 59-63) and in References (e.g. 12, 14). Check pleas punctuation, e.g. l. 55.

Essential oil could not be known as "compound" (l. 79) or "substance" (l. 97).

While dividing the table between pages the headline should be repeated.

Explain please the following questions applied to In vitro assay:

EOs main features are low water solubility and high volatility. It is hardly believable that 10% (and even 1%) solution of  EO  could be prepared in MEB or other water medium without solubilizing agents. Were plates protected against EO evaporation? How was MIC defined? What was the control?  Where are results of fungal viability test described in ls. 136-138?

Discuss please rather high MIC values obtained in your study.

Author Response

Dear Reviewer,

thank you for your kind revision.

Attached herewith you will find the answers to your concerns.

Very sincerely,

Simona Nardoni and co-authors

Round 2

Reviewer 2 Report

The manuscript is still not enough to publish as article. As we said in the previous review the studied EOs are chemically know as well as their antimicrobial activities are known. Moreover, there is only one test for one microorganism strain, even it is new but this is not enough to accept as article, may be as short report. However, the method of the preparations and obtaining of the EOs are unknown, and this makes the biological test not sure clear. The starting plant material must collected and prepared by authors, in any scientific research (e.g. you have no data about the used parts of the plants, that were used to obtain the EOs